# Probabilistic Active Meta-Learning

**Jean Kaddour**[*]
Department of Computer Science
University College London

**Steindór Sæmundsson**[*]
Department of Computing
Imperial College London

**Marc Peter Deisenroth**
Department of Computer Science
University College London

## Abstract

Data-efficient learning algorithms are essential in many practical applications where data collection is expensive, e.g., in robotics due to the wear and tear. To address this problem, meta-learning algorithms use prior experience about tasks to learn new, related tasks efficiently. Typically, a set of training tasks is assumed given or randomly chosen. However, this setting does not take into account the sequential nature that naturally arises when training a model from scratch in real-life: how do we collect a set of training tasks in a data-efficient manner? In this work, we introduce task selection based on prior experience into a meta-learning algorithm by conceptualizing the learner and the active meta-learning setting using a probabilistic latent variable model. We provide empirical evidence that our approach improves data-efficiency when compared to strong baselines on simulated robotic experiments.

## 1 Introduction

Learning models of complicated phenomena from scratch, using models with generic inductive biases, typically requires large datasets. Meta-learning addresses this problem by taking advantage of prior experience in a domain to learn new tasks efficiently. Meta-models capture global properties of the domain and use them as learned inductive biases for subsequent tasks. Standard in such algorithms is to randomly choose training tasks, e.g. by uniformly sampling parameterizations on the fly [1, 2].

However, exhaustively exploring the task domain is impractical in many real-world applications and uniform sampling is often sub-optimal [3]. For example, consider learning a meta-model of the dynamics of a robotic arm for a range of parameterizations, e.g., varying lengths and link weights. Due to costs, such as its wear and tear, there is a limited budget for experiments. Uniform sampling of the parameters/configurations, or even space-filling designs, may lead to uninformative tasks being explored due to the non-linear relationship between the parameters and the dynamics. In general, the relevant task parameters might not even be observed, rendering a direct search infeasible.

In this work, we adopt the view that the aim of a meta-learning algorithm is not only to learn a meta-model that generalizes quickly to new tasks, but to use its experience to inform which task is learned next. A similar view is found in Automatic curriculum learning (ACL) where, in general, a task selector is learned based on past data by optimizing it with respect to some performance and/or exploration metric [4]. For instance, the work in [5] uses automatic domain randomization to algorithmically generate task distributions of increasing difficulty, enabling generalization from simulation to real-life robots. Similarly motivated work is found in [6], referred to as unsupervised

---

[*]Equal contribution, correspondence to jean.kaddour.20@ucl.ac.uk

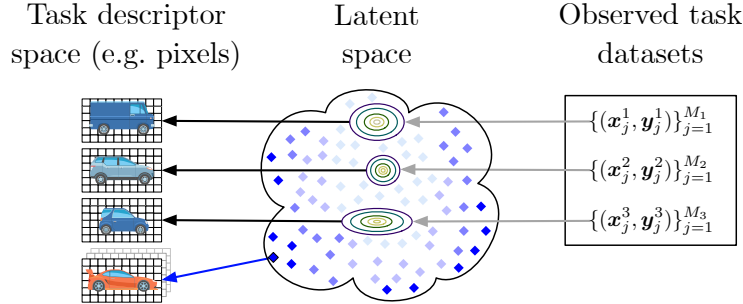

Figure 1: PAML infers latent embeddings of observed task datasets (Gaussian-shaped distributions, gray arrows), providing meaningful information about their relations and simultaneously learns a mapping to the task descriptor space (black arrows). It then ranks candidate tasks (diamonds) in the latent space based on their utility (the higher, the darker) and selects the one with highest utility.

meta-learning, and extended to ACL in [7]. Here, unsupervised pre-training is used to improve downstream performance on related RL tasks. In comparison to ACL, we note that our key objective is data-efficient exploration of a task space from scratch.

More closely related to our goal is active domain randomization in [3], which compares policy rollouts on potential reinforcement learning (RL) tasks compared to a reference environment, dedicating more time to tasks that cause the agent difficulties. PAML learns a representation of the space of tasks and makes comparisons directly in that space. This way our approach does not require a) rollouts on new potential tasks, b) handpicked reference tasks and c) the task parameters to be observed directly.

In contrast, we consider an unsupervised multi-modal setting, where we learn latent representations of task domains from *task descriptors* in addition to observations from individual tasks. A task descriptor might comprise (partially) observed task parameterizations, which is common in system configurations in robotics, molecular descriptors in drug design [8] or observation times in epidemiology [9]. In other cases, task descriptors might only indirectly contain information about the tasks, e.g., a grasping robot that can choose tasks based on images of objects but learns to grasp each object/task through tactile sensors. Importantly, the task descriptors resolve to a new task when selected.

Our main contribution is a probabilistic active meta-learning (PAML) algorithm that improves data-efficiency by selecting which tasks to learn next based on prior experience. The key idea is to use probabilistic latent task embeddings, illustrated in Figure 1, in a multi-modal approach to learn and quantify how tasks relate to each other. We then present an intuitive way to score potential tasks to learn next in latent space. Crucially, since the task embeddings are learned, ranking can be performed in a relatively low-dimensional space based on potentially complex high-dimensional data (e.g., images). Since the task-descriptors are made explicit in the model, additional interactions are not required to evaluate new tasks. PAML works well on a variety of challenging tasks and reduces the overall number of tasks required to explore and cover the task domain.

## 2 Probabilistic Meta-Learning

This section gives an overview of meta-learning models, focusing on probabilistic variants. We consider the supervised setting, but the exposition is largely applicable to other settings with the appropriate adjustments in the equations.

Meta-learning models deal with multiple task-specific datasets, i.e., tasks $\mathcal{T}_i$, $i = 1, \ldots, N$, give rise to observations $\mathcal{D}_{\mathcal{T}_i} = \{(\boldsymbol{x}_j^i, \boldsymbol{y}_j^i)\}$ of input-output pairs indexed by $j = 1, \ldots, M_i$. The tasks are assumed to be generated from an unknown task distribution $\mathcal{T}_i \sim p(\mathcal{T})$ and the data from an unknown conditional distribution $\mathcal{D}_{\mathcal{T}_i} \sim p(\mathbf{Y}^i | \mathbf{X}^i, \mathcal{T}_i)$, where we have collected data into matrices $\mathbf{X}^i, \mathbf{Y}^i$. The joint distribution over task $\mathcal{T}_i$ and data $\mathcal{D}_{\mathcal{T}_i}$ is then

$$p(\mathbf{Y}^i, \mathcal{T}_i | \mathbf{X}^i) = p(\mathbf{Y}^i | \mathcal{T}_i, \mathbf{X}^i) p(\mathcal{T}_i). \tag{1}$$

Generally speaking, we do not observe $\mathcal{T}_i$. Therefore, we model the task specification by means of a local (task-specific) latent variable, which is made distinct from global model parameters $\boldsymbol{\theta}$, which

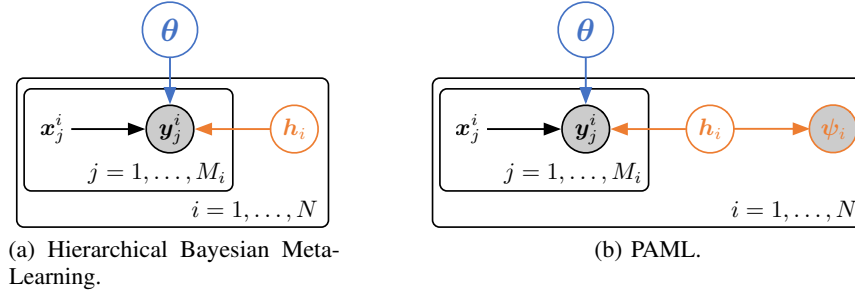

(a) Hierarchical Bayesian Meta-Learning.

(b) PAML.

Figure 2: Graphical models in the context of a supervised learning problem with inputs $x$ and targets $y$. Global parameters $\boldsymbol{\theta}$ (blue) are shared by all tasks, whereas local parameters $\boldsymbol{h}_i$ (orange) are specific to each task. (a) Hierarchical Bayesian Meta-Learning, e.g., [10, 11]. (b) PAML with additional task descriptors $\boldsymbol{\psi}_i$ that are conditioned on task-specific latent variables $\boldsymbol{h}_i$.

are shared among all tasks. Specifically, we follow Sæmundsson et al. [10] and learn a continuous latent representation $\boldsymbol{h}_i \in \mathbb{R}^Q$ of task $\mathcal{T}_i$. That is, we formulate the probabilistic model

$$p(\mathbf{Y}, \mathbf{H}, \boldsymbol{\theta}|\mathbf{X}) = \prod_{i=1}^{N} p(\boldsymbol{h}_i) \prod_{j=1}^{M_i} p(\boldsymbol{y}_j^i|\boldsymbol{x}_j^i, \boldsymbol{h}_i, \boldsymbol{\theta}) p(\boldsymbol{\theta}), \qquad (2)$$

where $\mathbf{H}$ collects the latent task variables. Global parameters $\boldsymbol{\theta}$ represent properties of the observations that are shared by all tasks, whereas each local task variable $\boldsymbol{h}_i$ models task-specific variation. For example, a family of sine waves $y(t) = A\sin(\omega t + \phi)$ parameterized by amplitude $A$, angular frequency $\omega$ and phase $\phi$ share the form of $y(t)$ (global) and have task specific parameters $A, \omega, \phi$ (local). Figure 2(a) shows the graphical model for the probabilistic model defined by (2). The likelihood $p(\boldsymbol{y}_j^i|\boldsymbol{x}_j^i, \boldsymbol{h}_i, \boldsymbol{\theta})$ factorizes given both the global parameters $\boldsymbol{\theta}$ and the local task variables $\boldsymbol{h}_i$.

Learning the model in (2) is intractable in most cases of interest, but is amenable to scalable approximate inference using stochastic variational inference. Alternatively, since the global model parameters $\boldsymbol{\theta}$ are estimated from all tasks, we can reasonably learn a point estimate using either maximum likelihood or maximum a posteriori estimation. To make this explicit in the exposition, we collapse the distribution over $\boldsymbol{\theta}$ and denote the model by $p_{\boldsymbol{\theta}}(\mathbf{Y}, \mathbf{H}|\mathbf{X}) = p_{\boldsymbol{\theta}}(\mathbf{Y}|\mathbf{H}, \mathbf{X}) p(\mathbf{H})$, where we additionally assume a fixed prior over the task variables $p(\mathbf{H})$. To approximate the posterior over task variables, we specify a mean-field variational posterior (with parameters $\boldsymbol{\phi}$)

$$p_{\boldsymbol{\theta}}(\mathbf{H}|\mathbf{Y}, \mathbf{X}) \approx q_{\boldsymbol{\phi}}(\mathbf{H}) = \prod_{i=1}^{N} q_{\boldsymbol{\phi}}(\boldsymbol{h}_i), \qquad (3)$$

which factorizes across tasks. The form of $q_{\boldsymbol{\phi}}(\cdot)$ is chosen, such that learning is made tractable. A typical choice is a Gaussian distribution. More expressive densities are possible using recent techniques developed around generative modeling and variational inference; see, e.g., [12, 13].

For learning the model parameters $\boldsymbol{\theta}$ and variational parameters $\boldsymbol{\phi}$, the intractability of the model evidence $p_{\boldsymbol{\theta}}(\mathbf{Y}|\mathbf{X})$ is finessed by maximizing a lower bound on the evidence (ELBO)

$$\log p_{\boldsymbol{\theta}}(\mathbf{Y}|\mathbf{X}) \geq \mathbb{E}_{q_{\boldsymbol{\phi}}(\mathbf{H})}\left[\log \frac{p_{\boldsymbol{\theta}}(\mathbf{Y}, \mathbf{H}|\mathbf{X})}{q_{\boldsymbol{\phi}}(\mathbf{H})}\right] = \mathbb{E}_{q_{\boldsymbol{\phi}}(\mathbf{H})}\left[\log p_{\boldsymbol{\theta}}(\mathbf{Y}|\mathbf{H}, \mathbf{X}) + \log \frac{p(\mathbf{H})}{q_{\boldsymbol{\phi}}(\mathbf{H})}\right] =: \mathcal{L}_{ML}(\boldsymbol{\theta}, \boldsymbol{\phi}), \qquad (4)$$

where Jensen's inequality is used to move the logarithm inside the expectation. When the likelihood of the model factorizes across data (such as in (2)), the bound in (4) consists of an expectation over a nested sum of likelihood and regularization terms, i.e.,

$$\mathcal{L}_{ML}(\boldsymbol{\theta}, \boldsymbol{\phi}) = \sum_{i=1}^{N} \sum_{j=1}^{M_i} \mathbb{E}_{q_{\boldsymbol{\phi}}(\boldsymbol{h}_i)}\left[\log p_{\boldsymbol{\theta}}(\boldsymbol{y}_j^i|\boldsymbol{x}_j^i, \boldsymbol{h}_i)\right] - \sum_{i=1}^{N} \mathbb{KL}\left[q_{\boldsymbol{\phi}}(\boldsymbol{h}_i)||p(\boldsymbol{h}_i))\right]. \qquad (5)$$

This objective can be evaluated using a Monte-Carlo estimate using samples $\boldsymbol{h}_i \sim q_{\boldsymbol{\phi}}(\boldsymbol{h}_i)$. The second term in (5) is the negative Kullback-Leibler divergence between the approximate posterior

**Algorithm 1** PAML

1: **input:** Task descriptors (distribution $p(\boldsymbol{\psi})$ or fixed set $\{\boldsymbol{\psi}_i\}_{i=1}^N$), active meta-learner $\{p_{\boldsymbol{\theta}}, q_{\boldsymbol{\phi}}\}$, utility function $u(\cdot)$ and $N_{\text{init}}$
2: Sample initial $\Psi_{\text{init}}$ and task datasets $\mathcal{D} = \mathcal{D}_{\text{init}}$
3: **while** meta-training **do**
4:     Train active meta-learning model $p_{\boldsymbol{\theta}}$ and infer task embeddings $q_{\boldsymbol{\phi}}(\mathbf{H})$ (see section 3.1)
5:     Select candidate $\boldsymbol{\psi}^*$ by ranking in latent space $\boldsymbol{\psi}^* = \text{argmax}_{\boldsymbol{h}_*}\, u(\boldsymbol{h}_*)$ (see section 3.2)
6:     Observe new task $\mathcal{D}_{\boldsymbol{\psi}^*} \sim p(\boldsymbol{y}|\boldsymbol{x}, \boldsymbol{\psi}^*)$
7:     Add new task to dataset $\mathcal{D} = \mathcal{D} \cup \mathcal{D}_{\boldsymbol{\psi}^*}$
8: **end while**

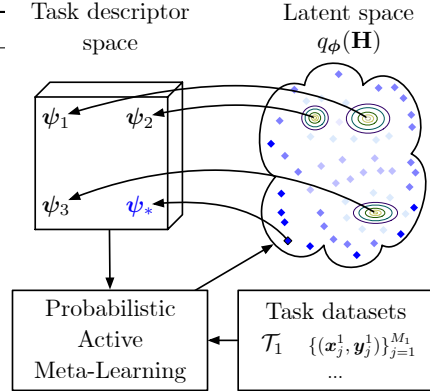

Figure 3: The Probabilistic Active Meta-Learning (PAML) Algorithm. PAML takes in a distribution or set of task descriptors from an underlying task domain $p(\mathcal{T})$, an active meta-learning model and a utility function. The task-descriptors $\boldsymbol{\psi}$, and observations $(\boldsymbol{x}, \boldsymbol{y})$, are used to learn latent embeddings $\boldsymbol{h}$ that model $\mathcal{T}$. PAML uses the latent embedding to do data-efficient active learning in task space.

$q_{\boldsymbol{\phi}}$ and the prior $p$ over latent task variables $\boldsymbol{h}_i$. When both $q_{\boldsymbol{\phi}}$ and $p$ are Gaussian, this term can be computed analytically. Since (5) consists of a sum over tasks $i$ and data $j$, we use stochastic gradient descent with mini-batches of data over both tasks and data within tasks to scale to large datasets.

At test time, we are faced with an unseen task $\mathcal{T}_*$, and our aim is to use the meta-model to make predictions $\mathbf{Y}_*$ given test inputs $\mathbf{X}_*$. A common scenario is a few-shot learning setting, where, given only a few data-points, we can perform predictions by approximate inference over the latent variable $q_{\boldsymbol{\phi}}(\boldsymbol{h}_*)$, keeping the model parameters fixed. Since the objective in (5) factorizes, we can efficiently optimize the variational parameters $\boldsymbol{\phi}$ of $q_{\boldsymbol{\phi}}(\boldsymbol{h}_*)$ given new observations only. Then, we make predictions using

$$p_{\boldsymbol{\theta}}(\mathbf{Y}_*|\mathbf{X}_*) = \mathbb{E}_{q_{\boldsymbol{\phi}}(\boldsymbol{h}_*)}\Big[ p_{\boldsymbol{\theta}}(\mathbf{Y}_*|\mathbf{X}_*, \boldsymbol{h}_*)\Big]. \tag{6}$$

Without any observations from the new task, we can make zero-shot predictions by replacing the variational posterior $q_{\boldsymbol{\phi}}(\boldsymbol{h}_*)$ in (6) with the prior $p(\boldsymbol{h}_*)$.

## 3 Probabilistic Active Meta-Learning

We are interested in actively exploring a given task domain in a setting where we have task-descriptive observations (task-descriptors), which we can use to select which task to learn next. In general, task-descriptors are any observations that enables discriminative inference about different tasks. For example, they might be fully or partially observed task parameterizations (e.g., weights of robot links), high-dimensional descriptors of tasks (e.g., image data of different objects for grasping), or simply a few observations from the task itself. Task-descriptors of task $\mathcal{T}_i$ are denoted by $\boldsymbol{\psi}_i$.

For active meta-learning, we require the algorithm to make either a discrete selection from a set of task-descriptors or to generate a valid continuous parameterization. In other words, the task-descriptors can be seen as actions available to the meta-model which transition it between tasks. From this perspective, the choice of task-descriptor (action-space) is either discrete or continuous and the task selection process can be seen as a restricted Markov decision process.

Figure 3 illustrates how PAML works. Given some initial experience $\mathcal{D}_{\text{init}}$, PAML trains the active meta-learning model from (7) (see Section 3.1) in steps 1–4. If the problem specifies a discrete set of candidates $\boldsymbol{\psi}_*$, we infer their corresponding latent variables $\boldsymbol{h}_*$ and rank them, see Section 3.2. Otherwise, we generate new candidates, e.g., by discretizing in latent space or sampling from the prior. These latent candidates are then used to generate new tasks $\boldsymbol{\psi}_*$, see (7). Finally, PAML observes the new task, adds it to the training set and repeats until a stopping criterion has been met (steps 6–8).

## 3.1 Extending the Meta-Learning Model

Our approach is based on the intuition that the latent embedding learned by the meta-learning model from Section 2 will, in some instances of interest, better represent differences between tasks than the task-descriptive observations on their own. Firstly, the latent embedding models the full source of variation due to task differences rather than using only partial information, as might be the case when there are hidden sources of task variation. Secondly, the embedding is both low dimensional and is required to explain variation in observations through the likelihood $p_{\boldsymbol{\theta}}(\boldsymbol{y}_j^i|\boldsymbol{x}_j^i, \boldsymbol{h}_i)$. If the task-descriptors contain redundant information, the model is implicitly encouraged to discard this in the latent embedding. To extend the meta-learning model in (2) to the active setting, we propose to learn the relationship between $\boldsymbol{h}_i$ and task-descriptors $\boldsymbol{\psi}_i$. Specifically, we propose the model

$$p_{\boldsymbol{\theta}}(\mathbf{Y}, \mathbf{H}, \boldsymbol{\Psi}|\mathbf{X}) = \prod_{i=1}^{N} p_{\boldsymbol{\theta}}(\boldsymbol{\psi}_i|\boldsymbol{h}_i)p(\boldsymbol{h}_i) \prod_{j=1}^{M_i} p_{\boldsymbol{\theta}}(\boldsymbol{y}_j^i|\boldsymbol{x}_j^i, \boldsymbol{h}_i), \tag{7}$$

where $\boldsymbol{\Psi}$ denotes a matrix of task-descriptive observations $\boldsymbol{\psi}_i$.

To train this model, we maximize a lower bound on the log-marginal likelihood

$$\log p_{\boldsymbol{\theta}}(\mathbf{Y}, \boldsymbol{\Psi}|\mathbf{X}) = \log \mathbb{E}_{q_{\boldsymbol{\phi}}(\mathbf{H})}\Big[p_{\boldsymbol{\theta}}(\mathbf{Y}|\mathbf{H}, \mathbf{X})p_{\boldsymbol{\theta}}(\boldsymbol{\Psi}|\mathbf{H})\frac{p(\mathbf{H})}{q_{\boldsymbol{\phi}}(\mathbf{H})}\Big] \tag{8}$$

$$\geq \mathbb{E}_{q_{\boldsymbol{\phi}}(\mathbf{H})}\Big[\log p_{\boldsymbol{\theta}}(\mathbf{Y}|\mathbf{H}, \mathbf{X}) + \log p_{\boldsymbol{\theta}}(\boldsymbol{\Psi}|\mathbf{H}) + \log \frac{p(\mathbf{H})}{q_{\boldsymbol{\phi}}(\mathbf{H})}\Big] \tag{9}$$

$$= \mathcal{L}_{ML}(\boldsymbol{\theta}, \boldsymbol{\phi}) + \sum_{i=1}^{N} \mathbb{E}_{q_{\boldsymbol{\phi}}(\boldsymbol{h}_i)}\big[\log p_{\boldsymbol{\theta}}(\boldsymbol{\psi}_i|\boldsymbol{h}_i)\big] =: \mathcal{L}_{PAML}(\boldsymbol{\theta}, \boldsymbol{\phi}), \tag{10}$$

where we used Jensen's inequality and a factorizing variational posterior $q_{\boldsymbol{\phi}}(\mathbf{H})$ as in (3).

By measuring the utility of a potential new task in latent space rather than through the task-descriptor $\psi$, the algorithm can take advantage of *learned* task similarities/differences that represents the *full* task configuration $\mathcal{T}$. The likelihood terms in equation (10), together with the prior on $\mathbf{H}$, means that two tasks that are similar are encouraged to be closer in latent space. Additionally learning the relationship between latent variables $h$ and $\psi$ provides a way of generating novel task-descriptors.

## 3.2 Ranking Candidates in Latent Space

A general way of quantifying the utility of a new task, in the context of efficient learning, is by considering the amount of information associated with observing a particular task [4]. To rank candidates in latent space, we define a mixture model using the approximate training task distribution $q_{\boldsymbol{\phi}}(\mathbf{H})$. We then define the utility of a candidate $\boldsymbol{h}_*$ as the self-information/surprisal [14] associated with $\boldsymbol{h}_*$, under this distribution:

$$u(\boldsymbol{h}_*) := -\log \sum_{i=1}^{N} q_{\boldsymbol{\phi}_i}(\boldsymbol{h}_*) + \log N. \tag{11}$$

When the approximate posterior $q_{\boldsymbol{\phi}_i}(\boldsymbol{h}_*)$ is an exponential family distribution, such as a Gaussian, equation (11) is easy to evaluate. We assign the same weight to each component because we assume the same importance for each observed task.

## 4 Experiments

In our experiments, we assess whether PAML speeds up learning task domains by learning a meta-model for the dynamics of simulated robotic systems. We test its performance on varying types of task-descriptors. Specifically, we generate tasks within domains by varying configuration parameters of the simulator, such as the masses and lengths of parts of the system. We then perform experiments where the learning algorithm observes: (i) fully observed task parameters, (ii) partially observed task parameters, (iii) noisy task parameters and (iv) high-dimensional image descriptors.

We compare PAML to uniform sampling (UNI), used in recent meta-learning work [1, 15] and equivalent to domain randomization [16], Latin hypercube sampling (LHS) of the parameterization

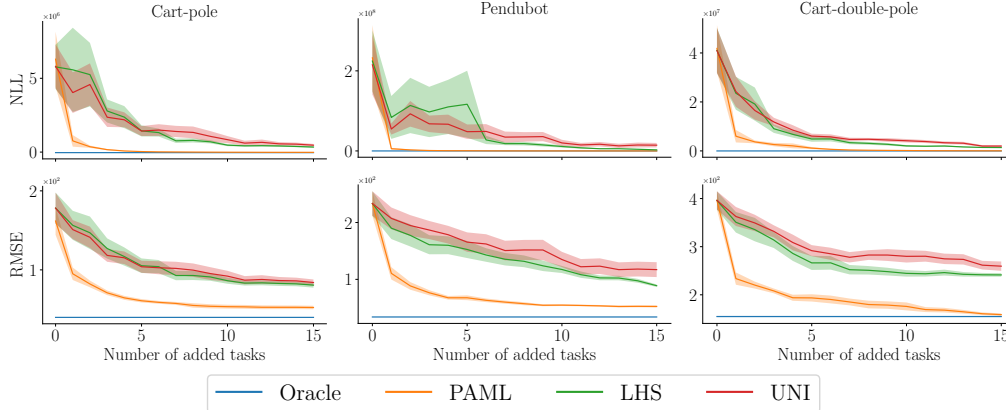

Figure 4: NLL/RMSE for 100 test tasks for the cart-pole, pendubot and cart-double-pole with observed task parameters as task-descriptors. Across all environments, PAML performs significantly better than the baselines UNI and LHS.

interval, and an oracle, i.e., the meta-learning model trained on the test tasks, representing an upper bound on the predictive performance given a fixed model. Fixed, evenly spaced grids of test task parameters are chosen to reasonably cover the task domain. As performance measures, we use the negative log-likelihood (NLL) as well as the root mean squared error (RMSE) on the test tasks. The NLL considers the full posterior predictive distribution at a test input, whereas the RMSE takes only the predictive mean into account. In all plots, error bars denote $\pm 1$ standard errors, across 10 randomly initialized trials.

We consider three robotic systems in the experiments, which are introduced below. The resulting dynamics models could also be used in model-based RL: the faster the model performs well in terms of predicting the task dynamics, the faster the planning algorithm will learn a good policy [17].

**Cart-pole** The cart-pole system consists of a cart that moves horizontally on a track with a freely swinging pendulum attached to it. The state of this non-linear system comprises the position and velocity of the cart as well as the angle and angular velocity of the pendulum. The control signals $u \in [-25, 25]$ N act as a horizontal force on the cart.

**Pendubot** The pendubot system is an underactuated two-link robotic arm. The inner link exerts a torque $u \in [-10, 10]$ Nm, but the outer joint cannot. The uncontrolled system is chaotic, so that modeling the dynamics is challenging. The system has four continuous state variables that consist of two joint angles and their corresponding joint velocities.

**Cart-double-pole** The cart-double-pole consists of a cart running on a horizontal track with a freely swinging double-pendulum attached to it. As in the cart-pole system, a horizontal force $u \in [-25, 25]$ N can be applied to the cart. The state of the system is the position and velocity of the cart as well as the angles and angular velocities of both attached pendulums.

Observations in these tasks consist of state-space observations, $\boldsymbol{x}, \dot{\boldsymbol{x}}$, i.e., position, velocity and control signals $\boldsymbol{u}$. We start with four initial tasks and then sequentially add 15 more tasks. To learn a dynamics model, we define the finite-difference outputs $\boldsymbol{y}_t = \boldsymbol{x}_{t+1} - \boldsymbol{x}_t$ as the regression targets. We use control signals that alternate back and forth from one end of the range to the other to generate trajectories. This policy resulted in better coverage of the state-space, compared to a random walk.

The meta-model learns a global function $\boldsymbol{y}_j^i = f_{\boldsymbol{\theta}}(\boldsymbol{x}_j^i, \boldsymbol{u}_j^i, \boldsymbol{h}_i)$ with local task-specific embeddings $\boldsymbol{h}_i$; see Section 2 for details. We choose to model the global function with a Gaussian process (GP) [18] as they are the gold standard for probabilistic regression. Specifically we use the sparse variational GP formulation from [19] and the meta-learning model developed in Section 3. The hyper-parameters of the GP play the role of the global parameters $\boldsymbol{\theta}$ and are shared across all tasks. A detailed description of (hyper-)parameters for the experiments is given in the Appendix.

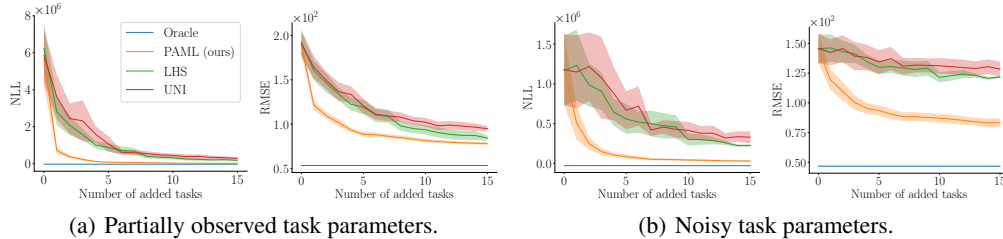

(a) Partially observed task parameters.       (b) Noisy task parameters.

Figure 5: NLL/RMSE for 100 test tasks for the cart-pole system with different task descriptors: (a) Partially observed task parameters; (b) Noisy task parameters. In all experiments, PAML performs significantly better than the baselines UNI and LHS.

## 4.1 Observed Task Parameters

In these experiments, the observed task descriptors match the task parameters exactly. However, the non-linear relationship between the parameters and the dynamics means that efficient exploration of the configuration space itself will, in general, not map directly to efficient exploration in terms of predictive performance. Here we test whether or not the meta-model learns latent embeddings that are useful for active learning of the task domain.

We specify task parameterization as follows: The cart-pole tasks differ by varying masses of the attached pendulum and the cart, $p_m \in [0.5, 5.0]$ kg and $p_l \in [0.5, 2.0]$ m, respectively. Pendubot and cart-double-pole tasks have lengths of both pendulums in the ranges, $p_{l_1}, p_{l_2} \in [0.6, 3.0]$ m and $p_{l_1}, p_{l_2} \in [0.5, 3.0]$ m, respectively.

Figure 4 shows the results of all methods in all three environments. Comparing PAML to the baselines UNI & LHS, we see that PAML performs significantly better than UNI and LHS in terms of performance on the test tasks. For all three systems, the NLL and RMSE see a steep initial drop for PAML, whereas the performance of the baselines drops more slowly and exhibits higher variance across experimental trials. This is because PAML consistently uses prior information to select the next task whereas the baselines are more affected by chance. We note that the gap in performance obtained by our approach over the baselines remains significant across the task horizon, which is particularly noticeable in the RMSE plots (bottom row) of Figure 4.

## 4.2 Partially Observed Task Parameters

Partial observability is a typical challenge when applying learning algorithms to real-world systems [20]. In these experiments, we simulate the cart-pole system where the task descriptors are chosen as the length of the pendulum, but we vary both its length and mass. In real life, one could imagine this scenario with space robots exposed to changing, unknown gravitational forces. The length is varied between $p_l \in [0.4, 3.0]$ m and the (unobserved) pendulum's mass $p_m \sim \mathcal{U}[0.4, 3.0]$ kg. I.e., each time a new task-descriptor is selected (i.e., length), the mass is sampled. In contrast, the oracle observes all possible masses $p_m$ within the test task grid. Results are shown in Figure 5(a). PAML achieves lower prediction errors in fewer trials than the baselines. The error after one added task of our methods is approximately matched by the baselines after about five added tasks. It selects similar lengths multiple times, which has the effect of exploring different values of the stochastic mass variable. For example, in one trial, the first eight selected lengths of PAML lie in the range $[0.41, 0.58]$ m. Intuitively, the reason for this is that the latent embedding represents the full task parameterization, and smaller values of the length make the effects of varying the mass more apparent. We interpret these results as a demonstration of how PAML is able to exploit information about unobserved task configuration parameters inferred by the meta-model.

## 4.3 Noisy Task Parameters

In this experiment, we explore the effects of adding a superfluous dimension to the task-descriptors. In particular, we simulate the cart-pole system where we add one dimension $\epsilon \in [0.5, 5.0]$ to the observations that does not affect the dynamics. To select tasks efficiently, PAML needs to learn to effectively ignore the superfluous dimension. Results in Figure 6 illustrate exactly this. Here we show

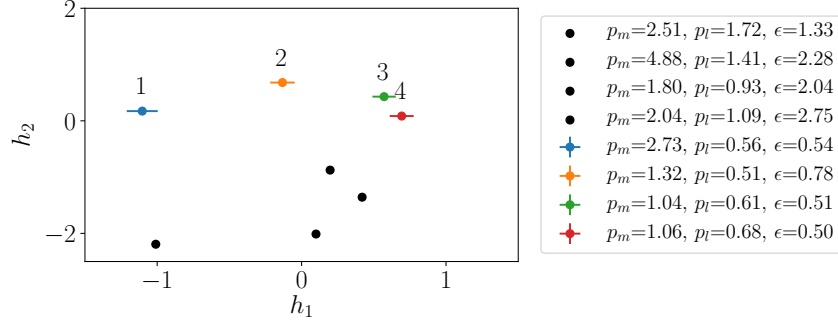

Figure 6: Latent embeddings from the cart-pole system with noisy task parmaeters. Black dots denote training tasks, and colored dots points chosen by PAML (with two standard deviation error bars). The numbers above each point denote the order they were picked.

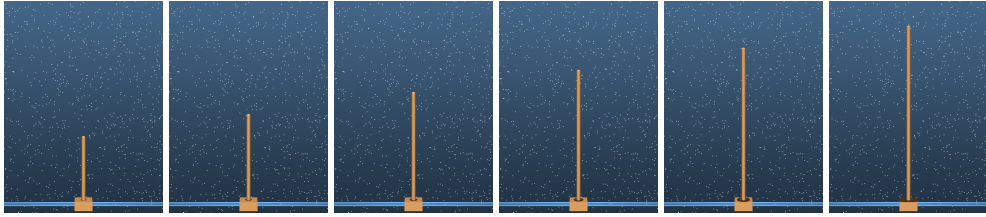

Figure 7: Pixel task-descriptors for the cart-pole system with different lengths. PAML can infer latent embeddings from pixel observations and exploit these for faster learning of a task domain.

the latent embeddings corresponding to the initial training tasks (black) and the selection made by PAML. We observe that it consistently picks a value for $\epsilon$ around $0.5$ while exploring informative values for $p_m$ and $p_l$. Figure 5(b) shows how predictive performance for PAML is better than the baselines in terms of both NLL and RMSE.

## 4.4 High-Dimensional (Pixel) Task Descriptors

In this experiment, PAML does not have access to the task parameters (e.g., length/mass) but observes indirect pixel task descriptors of a cart-pole system. We let PAML observe a single image of 100 tasks in their initial state (upright pole), where the pole length is varied between $p_l \in [0.5, 4.5]$. PAML selects the next task by choosing an image from this candidate set. The model then learns the dynamics of the corresponding task, from state observations $(x, \dot{x})$. We use a Variational Auto-Encoder [21, 22] to learn the latent variables from images (see Appendix for more details). Figure 7 shows example descriptors. The baseline selects images uniformly at random and both methods start with one randomly chosen training task. Figure 8 shows that PAML consistently selects more informative cart-pole images and approaches the oracle performance significantly faster than UNI.

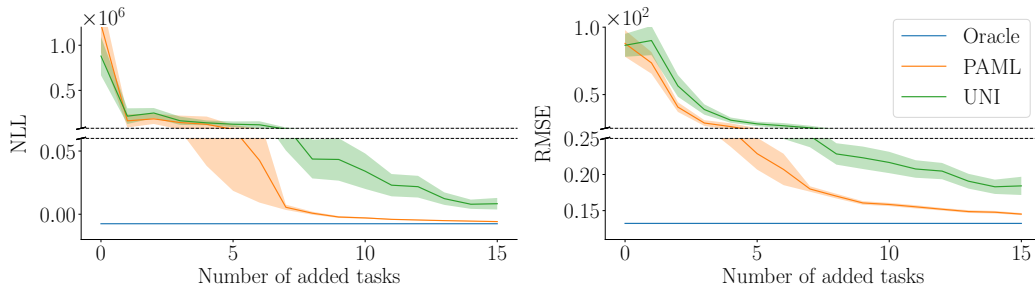

Figure 8: NLL/RMSE for 25 test tasks of the cart-pole system using pixel task-descriptors. PAML outperforms UNI by exploiting a learned latent representation of the task domain.

# 5 Conclusion

In this work, we proposed a general and data-efficient learning algorithm, combining ideas from active- and meta-learning. Our approach is based on the intuition that a class of probabilistic meta-learning models learn embeddings that can be used for faster learning. We extend ideas from meta-learning to incorporate task descriptors for active learning of a task domain, i.e., where the algorithm can choose which task to learn next by taking advantage of prior experience. Crucially, our approach takes advantage of learned latent task embeddings to find a meaningful space to express task similarities. We empirically validate our approach on learning challenging robotics simulations and show that it results in better performance than baselines while using less data.

## Broader Impact

The fundamental goal of this work is making learning algorithms more data-efficient. Fewer tasks to be observed might result in fewer experiments in real-world scenarios, directly reducing the resources needed to conduct these. Another consequence is shorter computation time during model training since less data is required. Less computation time reduces the overall energy consumption. Furthermore, the latent representation of tasks can be used to automatically infer similarities and commonalities between tasks, which may contribute to interpretability.

## Acknowledgments and Disclosure of Funding

S. Sæmundsson was supported by Microsoft Research through its PhD scholarship program. J. Kaddour thanks Stefan Leutenegger and Mark Hartenstein for fruitful discussions. We acknowledge a generous cloud credits award by Google Cloud.

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
