[Supplementary Material]

# A PAML with Gaussian processes

This section details how PAML can be combined with Gaussian processes, as in our experiments. Alternatively, one can use other probabilistic methods, e.g., Bayesian Neural Networks [1].

A Gaussian process is a probabilistic, non-parametric model and can be interpreted as a distribution over functions [2]. It is defined as an infinite collection of random variables $\{f_1, f_2, \dots\}$, any finite number of which are jointly Gaussian distributed. GPs are fully specified by a mean function $m$ and a covariance function (kernel) $k$, which allows us to encode high-level structural assumptions on the underlying function such as smoothness or periodicity.

Our mean function is specified by $m(\cdot) \equiv 0$ and we use the squared exponential (RBF) covariance function

$$k(\boldsymbol{x}_i, \boldsymbol{x}_j) = \sigma_f^2 \exp\left(-\tfrac{1}{2}(\boldsymbol{x}_i - \boldsymbol{x}_j)^\top \mathbf{L}^{-1}(\boldsymbol{x}_i - \boldsymbol{x}_j)\right), \tag{1}$$

where $\sigma_f^2$ is the signal variance and $\mathbf{L}$ is a diagonal matrix of squared length-scales. Each dimension of the targets $\boldsymbol{y}$ is modeled by an independent GP. A Gaussian likelihood is used and defined by

$$p(\boldsymbol{y}|\boldsymbol{x}, \boldsymbol{h}, \boldsymbol{f}(\cdot), \boldsymbol{\theta}) = \mathcal{N}\big(\boldsymbol{y}|\boldsymbol{f}(\boldsymbol{x}, \boldsymbol{h}), \mathbf{E}\big), \tag{2}$$

where $\boldsymbol{\theta} = \{\mathbf{E}, \mathbf{L}, \sigma_f^2, Q\}$ are the model hyper-parameters which consist of the diagonal signal noise matrix $\mathbf{E} := \mathrm{diag}(\sigma_1^2, \dots, \sigma_D^2)$, the diagonal squared length-scale matrix $\mathbf{L}$, and the dimension of the latent space $Q$ and $\boldsymbol{f}(\cdot) = \big(f^1(\cdot), \dots, f^D(\cdot)\big)$ denotes a multi-dimensional function. We place a standard-normal prior $\boldsymbol{h}_i \sim \mathcal{N}(\mathbf{0}, \mathbf{I})$ on the latent variables $\boldsymbol{h}_i$.

**Sparse variational GPs** Learning $N$ different tasks quickly becomes infeasible due to the $\mathcal{O}((MN)^3)$ computational complexity for training and $\mathcal{O}((NM)^2)$ for predictions, where $M$ is the number of data points per task. To address this issue, we turn to the sparse variational GP formulation from [3] and approximate the posterior GP with a variational distribution $q_\phi(\boldsymbol{f}(\cdot))$ which depends on a small set of $L \ll NM$ inducing points, where $NM$ is the total number of data points, given that we observe $M$ time steps for $N$ tasks. With a set of $L$ inducing inputs $\mathbf{Z} = (\boldsymbol{z}_1, \dots, \boldsymbol{z}_L) \in \mathbb{R}^{L \times (D+Q)}$ and corresponding GP function values $\mathbf{U} = (\boldsymbol{u}_1, \dots, \boldsymbol{u}_L) \in \mathbb{R}^{L \times D}$, we specify the variational approximation as a combination of the conditional GP prior and a variational distribution over the inducing function values,

$$q(f^d(\cdot)) = \int p(f^d(\cdot)|\boldsymbol{u}^d) q(\boldsymbol{u}^d)\, \mathrm{d}\boldsymbol{u}^d, \tag{3}$$

independently across all output dimensions $d$, where $q(\boldsymbol{u}^d) = \mathcal{N}(\boldsymbol{u}^d|\boldsymbol{m}^d, \mathbf{S}^d)$ is a full-rank Gaussian distribution. To optimize the variational parameters $\phi$ for the latent variables, we use a single sample $\boldsymbol{h}_i \sim q_\phi(\boldsymbol{h}_i)$ drawn from the variational distribution for each system that assumes independence between the latent functions of the GP $q_\phi(\boldsymbol{f}(\cdot))$ and the latent task variables

$$q_\phi(\boldsymbol{f}(\cdot), \mathbf{H}) = q_\phi(\boldsymbol{f}(\cdot)) q_\phi(\mathbf{H}). \tag{4}$$

We compute the integral in (3) in closed form since both terms are Gaussian, resulting in a GP with mean and covariance functions given by

$$m_q(\cdot) = \boldsymbol{k}_Z^\top(\cdot) \mathbf{K}_{ZZ}^{-1} \boldsymbol{m}^d, \tag{5}$$

$$k_q(\cdot, \cdot) = k(\cdot, \cdot) - \boldsymbol{k}_Z^\top(\cdot) \mathbf{K}_{ZZ}^{-1}(\mathbf{K}_{ZZ} - \mathbf{S}^d) \mathbf{K}_{ZZ}^{-1} \boldsymbol{k}_Z(\cdot) \tag{6}$$

with $[\boldsymbol{k}_Z(\cdot)]_i = k(\cdot, \boldsymbol{z}_i)$ and $[\mathbf{K}_{ZZ}]_{ij} = k(\boldsymbol{z}_i, \boldsymbol{z}_j)$. Here, the variational approach has two main benefits: Firstly, it reduces the complexity of training to $\mathcal{O}(NML^2)$ and predictions to $\mathcal{O}(NML)$. Secondly, it enables mini-batch training for further improvement in computational efficiency.

**Latent variables** For the latent variables $\mathbf{H}$, we assume a Gaussian variational posterior

$$q_\phi(\mathbf{H}) = \prod_{i=1}^{N} \mathcal{N}(\boldsymbol{h}_i|\boldsymbol{n}_i, \mathbf{T}_i), \tag{7}$$

where $\mathbf{T}_i$ is a full-rank covariance matrix. We use a diagonal covariance for more efficient computation of the ELBO. We obtain $\boldsymbol{h}_i$ by concatenating $\boldsymbol{n}_i$ and the diagonal covariance matrix entries of $\mathbf{T}_i$, which fully specifies the Gaussian latent variable.

**Evidence Lower Bound (ELBO)** The GP hyper-parameters $\boldsymbol{\theta}$ and the variational parameters $\boldsymbol{\phi} = \{\mathbf{Z}, \{\boldsymbol{m}_l, \mathbf{S}_l\}_{l=1}^{L}, \{\boldsymbol{n}_i, \mathbf{T}_i\}_{i=1}^{N}\}$ are jointly optimized when maximizing the ELBO. For training $p_{\boldsymbol{\theta}}(\mathbf{Y}, \mathbf{H}, \boldsymbol{f}(\cdot), \boldsymbol{\Psi} | \mathbf{X})$ w.r.t. $\boldsymbol{\theta}, \boldsymbol{\phi}$, we maximize the ELBO

$$\mathcal{L}_{PAML}(\boldsymbol{\Psi}, \boldsymbol{\theta}, \boldsymbol{\phi}) = \mathbb{E}_{q_{\boldsymbol{\phi}}(\boldsymbol{f}(\cdot), \mathbf{H})}\left[\log \frac{p_{\boldsymbol{\theta}}(\boldsymbol{\Psi}) p_{\boldsymbol{\theta}}(\mathbf{Y}, \mathbf{H}, \boldsymbol{f}(\cdot) | \mathbf{X})}{q_{\boldsymbol{\phi}}(\boldsymbol{f}(\cdot), \mathbf{H})}\right] \tag{8}$$

$$= \mathbb{E}_{q_{\boldsymbol{\phi}}(\boldsymbol{f}(\cdot), \mathbf{H})}\left[\log \frac{\prod_{i=1}^{N} p_{\boldsymbol{\theta}}(\boldsymbol{\psi}_i | \boldsymbol{h}_i) p_{\boldsymbol{\theta}}(\boldsymbol{h}_i) \prod_{j=1}^{M} p_{\boldsymbol{\theta}}(\boldsymbol{y}_j^i | \boldsymbol{x}_j^i, \boldsymbol{h}_i, \boldsymbol{f}(\cdot)) p_{\boldsymbol{\theta}}(\boldsymbol{f}(\cdot))}{q_{\boldsymbol{\phi}}(\boldsymbol{f}(\cdot), \mathbf{H})}\right] \tag{9}$$

$$= \sum_{i=1}^{N} \mathbb{E}_{q_{\boldsymbol{\phi}}(\boldsymbol{h}_i)}\left[\log p_{\boldsymbol{\theta}}(\boldsymbol{\psi}_i | \boldsymbol{h}_i)\right] + \sum_{i=1}^{N} \sum_{j=1}^{M} \mathbb{E}_{q_{\boldsymbol{\phi}}(\boldsymbol{f}_j^i | \boldsymbol{x}_j^i, \boldsymbol{h}_i) q_{\boldsymbol{\phi}}(\boldsymbol{h}_i)}\left[\log p_{\boldsymbol{\theta}}(\boldsymbol{y}_j^i | \boldsymbol{f}_j^i)\right]$$

$$- \mathbb{KL}(q_{\boldsymbol{\phi}}(\mathbf{H}) \parallel p_{\boldsymbol{\theta}}(\mathbf{H})) - \mathbb{KL}(q_{\boldsymbol{\phi}}(\boldsymbol{f}(\cdot)) \parallel p_{\boldsymbol{\theta}}(\boldsymbol{f}(\cdot))), \tag{10}$$

where we denote a collection of vectors in bold uppercase and we have dropped dependence on $\boldsymbol{\theta}, \boldsymbol{\phi}$ for notation purposes. We emphasize that $q_{\boldsymbol{\phi}}(\boldsymbol{f}_j^i | \boldsymbol{x}_j^i, \boldsymbol{h}_i)$ is the marginal distribution of the GP evaluated at the inputs $\boldsymbol{x}_j^i$. The KL term for the latent variables $\mathbb{KL}(q_{\boldsymbol{\phi}}(\mathbf{H}) \parallel p_{\boldsymbol{\theta}}(\mathbf{H}))$ is analytically tractable since both distributions are Gaussian. The KL term between the GPs $\mathbb{KL}(q_{\boldsymbol{\phi}}(\boldsymbol{f}(\cdot)) \parallel p_{\boldsymbol{\theta}}(\boldsymbol{f}(\cdot)))$ has been shown to simplify to $\mathbb{KL}(q_{\boldsymbol{\phi}}(\mathbf{U}) \parallel p_{\boldsymbol{\theta}}(\mathbf{U}))$ [4], which again is analytically tractable since both distributions are Gaussian. Thus, the ELBO can be written as

$$\mathcal{L}_{PAML} = \sum_{i=1}^{N} \mathbb{E}_{q_{\boldsymbol{\phi}}(\boldsymbol{h}_i)}\left[\log p_{\boldsymbol{\theta}}(\boldsymbol{\psi}_i | \boldsymbol{h}_i)\right] + \sum_{i=1}^{N} \sum_{j=1}^{M} \mathbb{E}_{q_{\boldsymbol{\phi}}(\boldsymbol{f}_j^i | \boldsymbol{x}_j^i, \boldsymbol{h}_i) q_{\boldsymbol{\phi}}(\boldsymbol{h}_i)}\left[\log p_{\boldsymbol{\theta}}(\boldsymbol{y}_j^i | \boldsymbol{f}_j^i)\right]$$

$$- \mathbb{KL}(q_{\boldsymbol{\phi}}(\mathbf{H}) \parallel p_{\boldsymbol{\theta}}(\mathbf{H})) - \mathbb{KL}(q_{\boldsymbol{\phi}}(\mathbf{U}) \parallel p_{\boldsymbol{\theta}}(\mathbf{U})). \tag{11}$$

The expected log-likelihood term needs further consideration: we would like to integrate out the latent variable $\boldsymbol{h}_i$ to obtain

$$q_{\boldsymbol{\phi}}(\boldsymbol{f}_j^i | \boldsymbol{x}_j^i) = \int q_{\boldsymbol{\phi}}(\boldsymbol{f}_j^i | \boldsymbol{x}_j^i, \boldsymbol{h}_i) q(\boldsymbol{h}_i) \, \mathrm{d}\boldsymbol{h}_i. \tag{12}$$

The integral in (12) is intractable due to the non-linear dependence on $\boldsymbol{h}_i$ in (5) and (6). Given our choice of the kernel function (RBF) and the fact that the likelihood $p(\mathbf{Y}|\boldsymbol{f})$ and the variational distribution $q(\boldsymbol{h}_i)$ are Gaussian, the first and second moments can be computed in closed form so that the log-likelihood term of the ELBO could be computed in closed form. However, instead of computing the first and second moments in closed form, we approximately integrate out the latent variable using Monte Carlo sampling for two reasons. Firstly, computing the moments can be prohibitively expensive since it requires the evaluation of a $NML^2D$ tensor. Secondly, computing the moments does not work for arbitrary kernel functions.

## B  Experimental details

**Observations** Observations consist of state-space observations, $\boldsymbol{x}, \dot{\boldsymbol{x}}$, i.e., position, velocity and control signals $\boldsymbol{u}$. We start with a small number of initial tasks and then sequentially add 15 more tasks. To learn a dynamics model, we define the finite-difference outputs $\boldsymbol{y}_t = \boldsymbol{x}_{t+1} - \boldsymbol{x}_t$ as the regression targets. During the evaluation, we compute the errors with respect to the normalized outputs, since the observed environments' state representations include dimensions of differing magnitudes, e.g., positions and velocities.

For generating the observations, we use the Deepmind Control Suite [5], powered by the MuJoCo Physics Engine [6]. Since the temporal integration is discrete with a fixed time-step $\Delta_t$ for all domains, we use the fourth-order Runge-Kutta method.

**Control signals** We use control signals that alternate back and forth from one end of the range to the other to generate trajectories. This policy resulted in better coverage of the state-space, compared to a random walk. The control signals are generated as an alternating sequence $\{\frac{C}{2}, \ldots, C, -\frac{C}{2}, \ldots, -C\}$, where $\{\frac{C}{2}, \ldots, C\}$ is one alternation with $\frac{T}{A}$ steps, $T$ the number of trajectory steps, $A$ the number of alternations and $C$ the lower/upper bound of the control signals. We use the same control signals for both training and test tasks. For illustration purposes, Figure 1 shows four cart-pole instances with differing configurations after three control signals have been applied.

| $p_m = 0.6\text{kg}, p_l = 0.5\text{m}$ | $p_m = 0.6\text{kg}, p_l = 1.0\text{m}$ | $p_m = 6.0\text{kg}, p_l = 0.5\text{m}$ | $p_m = 6.0\text{kg}, p_l = 1.0\text{m}$ |

Figure 1: Four differently configured cart-poles after the same three control signals.

| Experiment | (i) CP | (i) PB | (i) CDP | (ii) CP | (iii) CP | (iv) CP |
|---|---|---|---|---|---|---|
| **Observations** | | | | | | |
| Time discretization $\Delta_t$ | $0.125\,\text{s}$ | $0.05\,\text{s}$ | $0.05\,\text{s}$ | $0.125\,\text{s}$ | $0.125\,\text{s}$ | $0.125\,\text{s}$ |
| Dim. of state space | 4 | 4 | 6 | 4 | 4 | 4 |
| Dim. of action space | 1 | 1 | 1 | 1 | 1 | 1 |
| Dim. of observation space | 5 | 6 | 8 | 5 | 5 | 5 |
| Trajectory length in steps | 100 | 100 | 100 | 100 | 100 | 100 |
| Trajectory length in seconds | $12.5\,\text{s}$ | $5\,\text{s}$ | $5\,\text{s}$ | $12.5\,\text{s}$ | $12.5\,\text{s}$ | $12.5\,\text{s}$ |
| Control alternations | 10 | 5 | 10 | 10 | 10 | 10 |
| **Training** | | | | | | |
| Training steps | 5000 | 5000 | 7000 | 5000 | 5000 | 10000 |
| $N_{\text{init}}$ training tasks | 3 | 4 | 3 | 3 | 4 | 1 |
| **Evaluation** | | | | | | |
| Test tasks | 100 | 100 | 100 | 100 | 100 | 25 |
| Latent variable inference steps | 100 | 100 | 100 | 100 | 100 | 100 |

Table 1: Experimental (hyper-)parameters for (i) observed task paramaters of cart-pole (CP), cart-double-pole (CDP), pendubot (PB), (ii) partially observed task parameters of CP, (iii) noisy task parameters of CP and, (iv) high-dimensional pixel task descriptors of CP.

**Model/Training**    For training the MLGP, we use stochastic mini-batches, sampling a small number of trajectories and their associated latent variable at a time. Empirically, we found standardizing the inputs ($\boldsymbol{x}$) and outputs ($\boldsymbol{y}$) crucial for successful training of the model. For optimization, we use Adam [7] with default hyper-parameters: $\alpha = 10^{-2}, \beta_1 = 0.9, \beta_2 = 0.999, \epsilon = 10^{-8}$. We specify the latent space by $Q = 2$ latent dimensions. The sparse variational approximation of the true GP posterior uses 300 inducing points. Table B shows all remaining parameters for each experiment.

To illustrate the overall evaluation setup, in Figure 2 we show the one-step ahead prediction curves on eight different tasks of the fully-specified cart-pole environment after three initial training tasks have been learned and no tasks have been selected by PAML.

In Figure 3, we show plots of a final model (after 15 added tasks) for 8 different task specifications. For better readability, we plot the trajectory of each task separately.

**Reproducibility**    The attached code files include batch files that can be run to reproduce all results. Each trial of an experiment takes $\sim$ 60 minutes with one Nvidia Tesla V100 16GB GPU.

## B.1   Experiments (i)–(iii)

**Candidate set generation**    To score candidate tasks, we need points in latent space that then can be ranked. To generate such points, we discretize an interval $\mathbf{I} = I_1 \times \cdots \times I_Q$ in the $Q$-dimensional latent space, that contains the points $\mathbf{H}_* \subset \mathbb{R}^Q$. Furthermore, we remove candidates that map to task configurations that are outside the given interval $\mathbf{I}_{\boldsymbol{\Psi}}$, e.g., points that map to task

$p_m = 0.4\text{kg}, p_l = 0.98\text{m}$

$p_m = 0.69\text{kg}, p_l = 0.69\text{m}$

$p_m = 0.69\text{kg}, p_l = 1.56\text{m}$

$p_m = 0.98\text{kg}, p_l = 0.40\text{m}$

$p_m = 0.98\text{kg}, p_l = 0.98\text{m}$

$p_m = 1.27\text{kg}, p_l = 1.27\text{m}$

$p_m = 1.56\text{kg}, p_l = 0.69\text{m}$

$p_m = 2.71\text{kg}, p_l = 2.71\text{m}$

Figure 2: Prediction plots of various cart-pole tasks after three initial training tasks. $\theta, \dot{\theta}, x, \dot{x}$ denote the angle's position, angle's velocity, cart's position and cart's velocity, respectively. The error bars denote $\pm 2$ standard deviations of the predictive posterior distribution.

parameters with negative length/mass. To find good values for $\mathbf{I}$, we compute the minimum and maximum of the training task embeddings' means and added slack values $\xi_{\text{MIN}} = -10, \xi_{\text{MAX}} = 10$ to determine the endpoints for each latent dimension $d \in \{1, \dots, D\}$, e.g., the first interval endpoint $a_d = \min \mathbb{E}[q_\phi(\mathbf{H}_d)] + \xi_{\text{MIN}}$. We then discretize this interval with 100 grid points per latent dimension. In all experiments, we use $Q = 2$ and have $100^2$ candidates.

## B.2 Experiment (iv): High-dimensional Pixel Task Descriptors

In this experiment, PAML does not have access to the task parameters (e.g., length/mass) but observes indirect pixel task descriptors of a cart-pole system. We let PAML observe a single image of 100 tasks in their initial state (upright pole), where the pole length is varied between $p_l \in [0.5, 4.5]$. PAML selects the next task by choosing an image from this candidate set. The image gets transferred from the candidate descriptor set to the training task descriptors set. The model then learns the dynamics of the corresponding task, from state observations $(\boldsymbol{x}, \dot{\boldsymbol{x}})$. We use a Variational Auto-Encoder (VAE) [8, 9] to learn the latent variables from images. After each added task dataset, the VAE model parameters are reinitialized and optimized from scratch again. Thereby, it also decorrelates subsequent task selections as the final model performance is dependent on a particular initialization [10, 11].

**Model** Both the VAE's encoder and decoder consists of two fully-connected hidden layers with 200 hidden units each and leaky ReLU activation functions. The encoder computes the latent variable parameters $\phi_{\boldsymbol{h}_i | \boldsymbol{\psi}_i} = \{\boldsymbol{n}_i, \mathbf{T}_i\}$ conditioned on a cart-pole image $\boldsymbol{\psi}_i$. For ranking an image, the utility only considers the latent variable's mean. Furthermore, we add a likelihood-term

Figure 3: Plots of the learnt dynamics model for the cartpole environment on test tasks. $\theta, \dot{\theta}, x, \dot{x}$ denote the angle's position, angle's velocity, cart's position and cart's velocity, respectively. The figure shows the true data points (black discs) and the predictive distributions (blue) with $\pm 2$ standard deviations. The model generalizes well to the test tasks.

$p(\mathbf{\Psi}_{\text{candidates}}|\mathbf{H}_{\text{candidates}})$ to the training objective $\mathcal{L}_{PAML}$, where $\mathbf{\Psi}_{\text{candidates}}$ are all candidate task descriptors available. That means, at each training step, the training objective additionally considers a reconstruction loss for all candidate task descriptors. To train the model w.r.t. this loss, we use Adam [7] with hyper-parameters $\alpha = 0.002, \beta_1 = 0.9, \beta_2 = 0.999, \epsilon = 10^{-8}$.

Figure 4: The latent space (first column) and samples from the VAE prior (remaining columns) during training for 10 epochs (1k steps in each).