[Reviews · NeurIPS 2020]

Review 1

Summary and Contributions: This paper proposes a meta-learning approach that models tasks' latent embeddings that help to select the most informative tasks to learn next. The contribution of the paper is a probabilistic framework for active meta-learning which uses the learnt latent task embedding to rank tasks in the order of their informativeness. The method is evaluated by learning the environment dynamics for several in classical reinforcement learning environments.

Strengths: The grounding of the proposed approach looks sound to me. The model formulation is intuitive and does not make any overly restrictive assumptions. The experimental results are clearly communicated and show consistent improvement in the quality of the learnt dynamics model by the proposed method. I liked the experimental setting that includes fully observed task parameters, partially observed task parameters, noisy task parameters and pixel task parameters. This paper provides code for reproducing the experiment of the paper. While I didn't run the code, it looks to me well organized and reasonably documented.

Weaknesses: Empirical evaluation The method proposed in this paper learns the dynamics of the model that could be used as a model for model-based reinforcement learning. The evaluation is only done on the quality of the learnt model, but it would be very interesting to see how the improvement in the model translates into an improvement in the policy quality. The evaluation is done on 3 environments, but only one of the tasks is learnt from high-dimensional state space of images. It would be interesting to see such state-space in other tasks and also to see how the method could scale to other more complex tasks. Besides, the experimental evaluation contains some analysis/examples of the selected tasks in partial and noisy task parameters, but similar insights are missing in the results in fully observed and pixel task descriptors.

Correctness: The approach and the evaluation methodology look correct to me.

Clarity: The paper is very well written and illustrations are very well made. Some minor comments and questions: - At the very beginning of the article, it is not clear immediately what is meant by "task". It would help if it was introduced at the beginning of the introduction. - The illustrations are very nice, but unfortunately not readable in black and white. - "Noisy task parameters" makes me think more about adding some sampled noise to the task descriptors rather than adding a superfluous dimension. Maybe a more appropriate name could be related to "redundancy"? - It is not very clear to me in the setting with pixel task descriptors if the VAE embedding is used directly as a latent task embedding or not. If it is the case, does it mean that it is not jointly optimised with the meta-learning objective?

Relation to Prior Work: This paper discusses the prior work mostly in the introduction and related the proposed method with automatic curriculum learning, domain randomisation. In my opinion, the related work deserves a separate section in the article that would help to put the proposed method into the context. Besides, the state of meta-learning approaches could be discussed further. Finally, LHS method could be introduced in a bit more detail.

Reproducibility: Yes

Additional Feedback: - I did not find any reference in the text to Figure 2. - When learning dynamics of a particular task, the action control signal goes from one end of the range to another. How many actions are samples? I would be interested to see the extension that samples signals in the environment in an adaptive way as well. - It would be nice to have a figure for one of the environments that shows the quality of the learnt dynamic model from 4 different task specifications in the same plot. AFTER REBUTTAL: I would like to thank the authors for their response where they addressed the reviewers' concerns. I am mostly satisfied with the answers. However, I think that a more detailed discussion of the related work would be beneficial for the paper (as many of the reviewers suggested). In my assessment, I am still leaning towards the acceptance of the paper because the studded setting is interesting and the paper presentation is of high quality.


Review 2

Summary and Contributions: This paper proposes the importance of task selection in meta-learning and a variational method to select the next task to learn, exploiting task-descriptive observations / task-descriptors which are assumed available. Experimental results demonstrate the effectiveness and superiority of the proposed method.

Strengths: 1. Task selection is a valid problem in meta-learning, especially additional task-descriptors are available. 2. The proposed method is effective for task selection.

Weaknesses: 1. The technical novelty is limited.

Correctness: Yes

Clarity: Yes

Relation to Prior Work: 1. Discussion on related works is limited. No introduction for directions of related approaches. 2. Experimental comparisons for related works are limited. Except for the uniform sampling baseline and an estimated upper bound from training on the test set, only one method, LHS, is compared.

Reproducibility: Yes

Additional Feedback: My raised concerns have not been well addressed. Specifically, the concern of "No introduction for directions of related approaches." was misunderstood and not addressed.


Review 3

Summary and Contributions: The paper describes an approach of selecting new tasks to learn in a meta-learning setting. They use a latent variable space to project tasks and rank them in this space. The paper mainly focusses on applications in robot control systems, with one of their experiments observing the state of the systems as images.

Strengths: The work is relevant to our community. Their claims are sound. The authors provide theoretical groundwork to establish their approach and provide empirical evaluations. The paper uses prior work of modeling tasks in the latent space by Sæmundsson et al and applies it to the meta-learning framework.

Weaknesses: The latent space formulation they have used for empirical evaluation can be easily applied to the chosen applications (robot control systems). However it is not clear if this approach can be extended to other domains. The paper either needs a discussion on this topic or it can choose to be written in the context of their domain.

Correctness: The claims and methods in this method are correct. They also provided code to run their experiments.

Clarity: The paper is well written.

Relation to Prior Work: The paper clearly describes prior work, but doesn't provide enough discussion on how it is different. I think this paper deserves a lot more discussion on prior work because of the nature of the domain and given that this community is not primarily a robotics community. The paper would be better if it expands on prior work and how it differentiates itself - - how is it different from optimal control? - what is the objective function of a robotic control task and what are they tasks this paper has chosen optimizes for? - why should someone choose meta learning rather than transfer learning for learning a control policy?

Reproducibility: Yes

Additional Feedback: Your work is most similar to other work in the robotics literature where a control task is being optimized. For example in Active Domain Randomization Mehta et al. use an acquisition function that optimizes a maximum utility function. The same with Gupta et. als work in Unsupervised Meta-Learning for RL (reaching the goal vs reaching the goal in the most optimal way). It is not clear to me which objective these meta learning process is optimizing for? Can you compare your work empirically closely with ACL? Thanks for documenting the code. LGTM! After Rebuttal: I am happy with the explanation and distinction between PAML, DR, and ACL. I think this paper should be accepted.


Review 4

Summary and Contributions: The authors introduce a probabilistic framework for active learning in meta-learning (PAML). In particular, they learn a latent representation for the tasks which is used to rank the utility of a new task. The main contribution is the addition of active learning to the meta-learning framework (and in particular, probabilistic meta-learning), which allows for efficient task selection rather than simply learning from a suite of tasks. This framework also allows for task creation when the family of tasks can be characterized by a continuous descriptor. Overall I think this paper is a good contribution, but I'm unclear on whether existing approaches can be applied to the experimental setup (see Weaknesses section below); if not, then I can raise the score, but if so, then the lack of experimental comparison is a serious drawback.

Strengths: This paper is novel, as a first attempt to do active learning for task selection in meta-learning. As the authors explain, labeling new tasks is expensive, and using active learning in this field seems like a valuable contribution. The authors explain clearly the background (probabilistic meta learning) and their design is sound. Their experiments use reasonable baselines and highlight that PAML leads to a better learning curve when incorporating new tasks.

Weaknesses: I’m not clear on whether existing techniques could be applied to the experimental setup. Specifically, I don’t see why other referenced techniques (e.g. Active Domain Randomization [3], some of what’s cited in [4], or regular Domain Randomization) can’t be used for some or possibly all of the experimental setups. The authors say that [3] “requires selecting reference parameters beforehand and direct access to the task parameters of interest,” but don’t the experiments in 4.1 (Observed Task Parameters) meet that criteria? And for 4.2 (Partially Obesrved) and 4.3 (Noisy), couldn’t [3] be applied to the observed parameters? If so, this is an important comparison and it is a drawback to not have it.

Correctness: The approach looks like a reasonable extension from prior work.

Clarity: The paper is clear and well-written in general, though I had two issues regarding clarity: the authors should be clearer on their contributions in relation to prior work (see comments in the Prior Work section below), and more importantly, I wasn’t clear on why existing techniques could not be applied to the various setups in the experimental section (see comments in the Weaknesses section above). On a more detailed level, I was confused by two different points in the paper: * The authors should explain in a few words what the LHS baseline is. * In the part 3.2, it would have been helpful to explain how you get h_i from the task observable characteristics (“we infer their corresponding latent variables”).

Relation to Prior Work: The authors should more clearly state what is existing work and what their novel contributions are. For example, the entirety of Section 2 on Probabilistic Meta-Learning seems to be prior work, but has no references. I believe that incorporating Active Learning to the Meta-Learning framework is the paper’s main contribution, but this should be clearly stated in the beginning of the paper.

Reproducibility: Yes

Additional Feedback: Is the uniform sampling (UNI) baseline the same as Domain Randomization? If so, this should be noted and directly cited (it’s currently indirectly cited through [3]). If not, the authors should explain whether or not it could be applied to this setup (similar to comments above).

[Author Response · NeurIPS 2020]

We highly appreciate the reviewers' time, efforts, and valuable suggestions! Recap: The contribution of our work is that we introduce task selection based on prior experience into a meta-learning algorithm by conceptualizing the learner and the active meta-learning setting using a probabilistic latent variable model. We are encouraged that reviewers find our approach intuitive without any overly restrictive assumptions [R1], effective for task selection [R2], relevant to our community with sound claims [R3], and novel as well as a valuable contribution to the field of meta-learning [R4].

**R2, R3, R4**  Comparisons with related work. Reviewers asked whether existing approaches, namely domain randomization (DR), active DR (ADR) [3] and active curriculum learning (ACL) [4,6,7], could be evaluated in the same setting. R3, R4 asked for further clarification on the differences between existing work and our approach. DR is equivalent to the UNI baseline in our experiments and we will add the appropriate citations. ADR is not applicable to our setup due to the kind of proxy that is used to rank the informativeness of a new task. ADR compares policy rollouts on potential tasks compared to a reference environment, dedicating more time to tasks that cause the agent difficulties. PAML learns a representation of the space of tasks and makes comparisons directly in that space. This way our approach does not require a) rollouts on new potential tasks, b) handpicked reference tasks and c) the task parameters to be observed directly. In comparison to ACL, we note that our key objective is data-efficient exploration of a task space from scratch. ACL performs unsupervised pre-training on environments to improve downstream RL tasks, making a direct comparison unsuitable. PAML and ACL can be seen as complimentary approaches, e.g., PAML might be used to select the tasks used for unsupervised pre-training in ACL.

**R1**  Empirical evaluation. We agree that it would be interesting to see how our approach would impact the quality of a policy in an RL setting, but see this as beyond the scope of the current work. We look at learning models, which can be used in a model-based RL setting, but additional (confounding) factors make policies successful, e.g., exploration and local optima. R1 also mentions that only one of the environments is learned from pixel data. With space constraints in mind, we chose to focus on evaluating PAML across the different task parameter scenarios where PAML is applicable. Lastly, we will add an analysis of the settings fully observed 4.1 and pixel-descriptor 4.4. to the paper: in 4.1, PAML consistently picks masses/lengths of the lower value end of the range but more diversely than in 4.2. In 4.4, it usually starts to select lengths at both ends of the range and then selects tasks towards the middle. VAE embedding. The VAE embedding is used directly as a latent task embedding and is jointly optimized with the meta-learning objective $\mathcal{L} = \mathcal{L}_{PAML} + p(\boldsymbol{\Psi}_{\text{candidates}}|\mathbf{H}_{\text{candidates}})$, where $\boldsymbol{\Psi}_{\text{candidates}}$ are candidate task pixel descriptors, see B.2 in the appendix. Discussion of meta-learning (ML). With space constraints in mind and since our work's goal is to incorporate active learning into ML rather than deriving a new ML method, we kept the part about prior work in ML short but detailed the ML approach used in this work in Section 2. Control signals. The number of actions is in Appendix Table 1.Plots of the model. In Figure 1, we show plots of the learnt model for 8 different task specifications. For better readability, we plot the trajectory of each task separately.

| $p_m = 0.5$kg, $p_l = 0.5$m | $p_m = 0.5$kg, $p_l = 2.0$m | $p_m = 1.0$kg, $p_l = 1.0$m | $p_m = 1.0$kg, $p_l = 2.0$m | $p_m = 2.0$kg, $p_l = 1.0$m | $p_m = 2.0$kg, $p_l = 2.0$m | $p_m = 5.0$kg, $p_l = 0.5$m | $p_m = 5.0$kg, $p_l = 2.0$m |

Figure 1: Plots of the learnt dynamics model for the cartpole environment on test tasks. $\theta, \dot{\theta}, x, \dot{x}$ denote the angle's position, angle's velocity, cart's position and cart's velocity, respectively. The figure shows the true data points (black discs) and the predictive distributions (blue) with $\pm 2$ standard deviations. The model generalizes well to the test tasks.

**R2**  Limited technical novelty. Although our work is built on existing ideas for probabilistic meta-learning and active learning, as far as we are aware, our algorithm addresses a gap in the literature when it comes to the combination of the two. Particularly in how we conceptualize task parameters in PAML's model. Our experiments show that this can be practically exploited in various scenarios. No introduction of related approaches. We introduced previous work (DR, ACL, ADR) that is similarly motivated and most close to ours in the third and fourth paragraph in Section 1.

**R3**  Domains outside robotic systems. Generally speaking, in situations where the meta-learning model is able to learn a useful task embedding our approach provides a way of taking advantage of that embedding to make informed task selections. Since data-efficient learning of task spaces is an important problem in robotics, we think that our selection of experiments does not detract from the evaluation but illustrates task conditions that are easy to interpret, such as the under-/ or over-specification.

[Meta-Review · NeurIPS 2020]

The reviewers agree that the paper addresses an interesting problem of active meta-learning and provides a novel solution.